# Changing Spring Phenology of Northeast China Forests during Rapid Warming and Short-Term Slowdown Periods

**Fengyuan Zhang** [1], **Binhui Liu** [1,*] , **Mark Henderson** [2] , **Xiangjin Shen** [3] , **Yuanhang Su** [1] **and Wanying Zhou** [1]

1   College of Forestry, The Northeast Forestry University, Harbin 150040, China
2   Mills College at Northeastern University, Oakland, CA 94613, USA
3   Northeast Institute of Geography and Agroecology, Chinese Academy of Sciences, Changchun 130102, China
*   Correspondence: binhui@nefu.edu.cn

**Abstract:** The vast forests of Northeast China are under great pressure from climate change. Understanding the effects of changing climate conditions on spring phenology is of great significance to assessing the stability of regional terrestrial ecosystems. Using Normalized Difference Vegetation Index data from 1982 to 2013, this paper investigated the changes in the start date of the vegetation growing season (SOS) of two main forest types in Northeast China, analyzing the changes in temporal and spatial patterns of forest spring phenology before and during the recent short-term warming slowdown, and exploring the effects of day and night temperatures and precipitation on the start of the growing season. The results showed that, during the rapid warming period (1982–1998), the SOS of deciduous needleleaf forests (DNF) was significantly advanced ($-0.428$ days/a, $p < 0.05$), while the rate of advance of SOS of deciduous broadleaf forests (DBF) was nonsignificant ($-0.313$ days/a, $p > 0.10$). However, during the short-term slowdown (1998–2013), the SOS of DBF was strongly delayed (0.491 days/a, $p < 0.10$), while the change in SOS of DNF was not significant (0.169 days/a, $p > 0.10$). The SOS was sensitive to spring maximum temperature for both forest types during the analysis period. Increases in winter precipitation influenced the SOS during the rapid warming period for DNF; this combined with the increase in the spring maximum temperature contributed to the advance in SOS. The decrease in the spring maximum temperature during the short-term slowdown, combined with a decrease in the previous summer maximum temperature, contributed to the rapid delay of SOS for DBF. DBF SOS was also more influenced by lagged effects of prior conditions, such as previous autumn to spring precipitation during the rapid warming period and previous summer maximum temperature during the short-term slowdown. In general, SOS was mainly determined by changes in daytime thermal conditions; DNF is more sensitive to temperature increases and DBF is more sensitive to decreases. Different regional climate conditions lead to differences in the distribution of DNF and DBF, as well as in the response of SOS to climate change during the rapid warming and short-term slowdown periods.

**Keywords:** short-term slowdown; spring phenology; Northeast China forests; NDVI; climate change

## 1. Introduction

As an important component of terrestrial ecosystems, forests play an important role in atmospheric regulation, material circulation, soil and water conservation, and biodiversity. The intensification of climate change in the past 50 years has affected the timing of vegetation growth in many ecosystems, including forests [1,2]. Exploring the impacts of climate change on forest growth and the potential effects of forest growth on climate have attracted increasing attention in the field of global ecology [3,4].

Spring phenology and, specifically, the starting date of the vegetation growing season (Start of Season, or SOS) are widely studied as important evaluative indicators of vegetation growth and development [5,6]. Changes in spring phenology can often change the carbon cycle and terrestrial productivity, with potential impacts on the stability of terrestrial

ecosystems [7]. Therefore, accurately monitoring the spatial and temporal changes of SOS is essential to understanding the response of terrestrial ecosystems to climate change.

Extensive studies around the world have found that global warming has significantly advanced SOS in the Northern Hemisphere, with important impacts on ecosystem cycles [8–10]. In recent years, more attention has been paid to the asymmetric effects of daytime and nighttime temperatures on SOS. Some previous studies have found that the SOS in high latitudes of the Northern Hemisphere is mainly driven by the maximum temperature [5,11]. However, Shen et al. (2016) [12] pointed out that the SOS of vegetation on the Qinghai–Tibet Plateau is mainly controlled by the minimum temperature. Another study showed that the SOS in China's temperate grasslands is mainly controlled by the minimum temperature in spring and the maximum temperature in winter [13]. Precipitation also has an important impact on SOS: the more arid the region, the more sensitive SOS is to precipitation [13–15]. These spatiotemporal differences must be accounted for in assessing different influences on the start of the vegetation growing season.

While global temperatures have risen in recent decades, the rate of increase in air temperatures in many instances slowed from 1998 to 2012, a phenomenon dubbed the "short-term slowdown" [16–18]. The effects of the short-term slowdown on SOS are unclear: while Wang et al. (2016) [19] found that SOS in the Northern Hemisphere advanced at a faster rate from 2000 to 2012 than in the 1980s, a subsequent study failed to detect a significant trend for the period of 1998 to 2014 [20]. These studies also noted strong spatial heterogeneity in SOS. More research is needed to explore the impact of warming rates on vegetation SOS [21].

Vegetation indices derived from remote sensing imagery have proven useful in systematically monitoring vegetation phenology [10,22]. By comparing vegetation indices from multiple points in time, we can document changes in phenology from year to year [23,24]. This method overcomes the shortcomings of ground surveys, which are time-consuming and laborious, and has the advantages of covering long time spans and wide geographical areas with high precision [25]. The Normalized Difference Vegetation Index (NDVI) is widely used as an indicator to monitor vegetation growth [26–28] and using it to reconstruct and extract vegetation phenology information is the mainstream technical means to obtain vegetation phenology by remote sensing [25,29].

The forests of northeast China, spanning latitudes from ~40° to 53° N, form an important ecological barrier and forestry base, rich in species diversity and sensitive to climate change. Studies have analyzed the changes of SOS in Northeast China, but at present there is little information about whether the relative impacts of daytime and nighttime temperatures affect SOS, and how SOS trends may have changed through the global short-term slowdown is also rarely reported.

In this study, we used the GIMMS NDVI dataset from 1982 to 2013 to monitor the changes in SOS of the major types of forests in Northeast China that did not undergo land cover change in this period, exploring the relative impacts of daytime and nighttime temperatures and precipitation on forest SOS and revealing the impact of the short-term slowdown. The purpose of this paper is to understand the response of forest ecosystems in middle and high latitudes to climate change, and to provide a scientific basis for regional forestry ecological management under changing climate conditions.

## 2. Materials and Methods

### 2.1. Study Area

The study area (see Figure 1) is located in the Northeast China provinces of Heilongjiang, Jilin, and Liaoning and the Inner Mongolia Autonomous Region, with a total area of $127 \times 10^4$ km² spanning 38°72′~53°55′ N to 115°52′~135°09′ E and ranging in altitude from 0 to 2667 m [30]. This area encompasses the largest natural forests in China and is an important forest production base. Most of the region belongs to the temperate continental monsoon climate zone, with distinct seasons; cold, dry winters; warm, humid summers; and decreasing annual rainfall gradients from east to west.

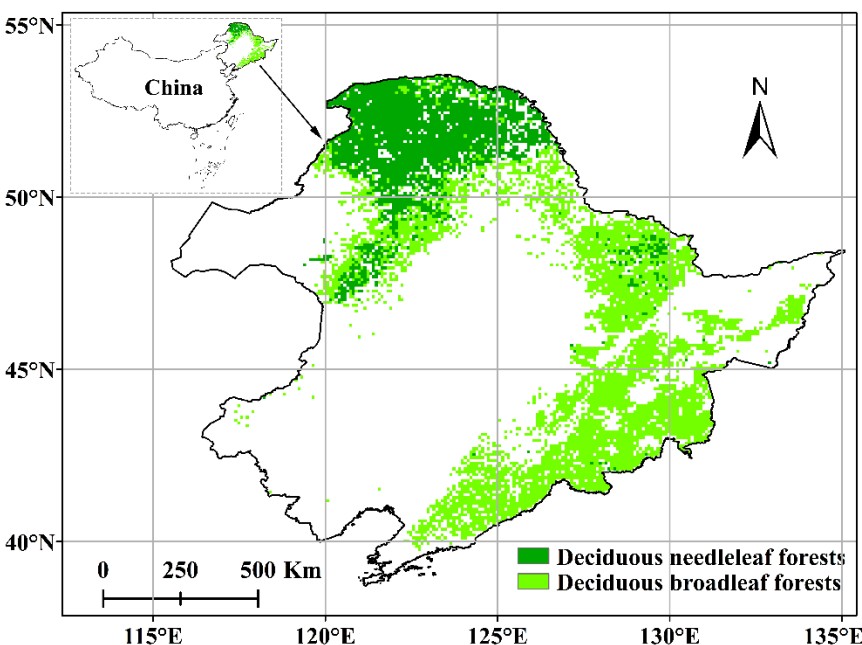

**Figure 1.** Location of study area and distribution range of two forest types.

We delineated the study area using the annual land cover datasets of the European Space Agency (ESA) climate change initiative (CCI) (http://maps.elie.ucl.ac.be/CCI/viewer/download.php, accessed on 9 January 2022). The dataset has a spatial resolution of 300 m and an estimated classification accuracy of over 70%. In order to reduce the impact of land cover change on the research results [31], we compared land cover for 1992 and 2015 and excluded pixels that changed between those dates. We then selected the two main types of forest vegetation, deciduous needleleaf forest (DNF) and deciduous broadleaf forest (DBF), which are widely distributed in Northeast China. The DNF and DBF forest types display obvious seasonal characteristics, making them conducive to the study of spring phenology [10]. The two forest types have obvious zonal characteristics: DNF is mainly found in the cold temperate zone around the Great Xing'an (Khingan) Mountains, while DBF is widely distributed south of the Great Xing'an, Lesser Xing'an, and Changbai Mountains in areas that are more humid with moderate temperature and precipitation. (Some other forest types are found in Northeast China, but they cover a relatively small area, so they were excluded from this study.)

*2.2. Data*

The remote sensing data used in this study came from the latest version of NDVI data (GIMMS NDVI 3g. v1) provided by the third generation Global Inventory Monitoring and Modeling System. The temporal resolution is 15 days. This data set is one of the NDVI data sets with the longest timespans and the highest utilization to date. It has been widely used in investigating vegetation change all over the world. The dataset has been corrected for pollution and noise to greatly reduce errors and spurious deviations [13,32]. To reduce the impact of bare ground and other noise on our analysis of NDVI timing, we excluded pixels with annual average NDVI values less than 0.1 [10,33]. The use of multiple datasets compiled at different scales necessarily introduces some degree of uncertainty; however, we took a conservative approach by aggregating the ESA land cover data (300 m spatial resolution) for analysis at the 8 km spatial resolution of the GIMMS NDVI dataset.

For meteorological data, this study relied on the CRUTS v4.05 dataset provided by the University of East Anglia, United Kingdom (https://crudata.uea.ac.uk/cru/data/hrg, accessed on 29 March 2022), with a spatial resolution of 0.5° and a time span from 1901 to 2020. We obtained the monthly average maximum temperature (Tmax), monthly average minimum temperature (Tmin), and monthly total precipitation (PCP) for grid cells

overlapping our study area, then resampled the meteorological data to be consistent with the 8km spatial resolution of the NDVI data [34]. We then computed seasonal values for each year—that is, summer (June to August of the previous year), autumn (September to November of the previous year), winter (December of the previous year to February of the current year), and spring (March to May of the current year). The short-term slowdown ended in 2012 but considering that SOS is often related to the climatic conditions of the previous year, the time span of data examined in this study is 1982–2013.

*2.3. Methods*

In this study, we used the double logistic-relative threshold method [29,35] to extract the Northeast Forest SOS date for each year. The principle is to use a double logistic function to fit NDVI data:

$$NDVI_{(t)} = a + b\left(\frac{1}{1 + e^{c(t-d)}} + \frac{1}{1 + e^{e(t-f)}}\right), \tag{1}$$

where $NDVI_{(t)}$ is the NDVI fitted on day t, a is the initial background NDVI value, and $a + b$ is the maximum NDVI value, d is the increasing inflection point (spring date), c is related to the rate of increase at the d inflection point, f is the decreasing inflection point (autumn date), e is related to the rate of decrease at the f inflection point. The fitted NDVI data were then normalized by:

$$NDVI_{ratio} = \frac{(NDVI - NDVI_{min})}{(NDVI_{max} - NDVI_{min})}, \tag{2}$$

where NDVI represents the daily NDVI data fitted with double logic function, $NDVI_{min}$ represents the minimum NDVI of each year, and $NDVI_{max}$ represents the maximum NDVI of each year. Following Wang et al. (2019) [20] and Zeng et al. (2021) [4], we identified SOS as the day when $NDVI_{ratio}$ increased to 0.2 in the spring.

We used simple linear regression to detect the trend of change in SOS in each period [1,31]. The formula for simple linear regression is:

$$\theta_{slope} = \frac{(n \times \sum_{i=1}^{n} i \times SOS_i) - (\sum_{i=1}^{n} i \sum_{i=1}^{n} SOS_i)}{n \times \sum_{i=1}^{n} i^2 - (\sum_{i=1}^{n} i)^2}, \tag{3}$$

where $\theta_{slope}$ represents the change trend (slope) of SOS of each pixel; n is the number of monitoring years; i is the year serial number; and $SOS_i$ is the SOS value of year i. If $\theta_{slope}$ were positive, that would indicate that SOS is delaying to later dates in the year; a negative slope would indicate that SOS is advancing to earlier dates in the year.

We used partial correlation analysis to calculate the correlation between seasonal climate factors and forest SOS in each period. Partial correlation can better solve the collinearity problem between climate variables and excludes the interference of other factors when analyzing the relationships between a certain climate factor and SOS [10,36]. The *t*-test was used to test the significance of the partial correlation coefficient.

## 3. Results

*3.1. Spatiotemporal Change of Spring Phenology*

We calculated the spatial distribution of SOS of the two main types of forests in the Northeast China study area from 1982 to 2013 (Figure 2). The average SOS of the forests mostly fall between DOY (day of the year) 105 and DOY 125 (from 15 April to 5 May, or from 14 April to 4 May in leap years). The average SOS of DNF is DOY 118.69, and that of DBF is DOY 119.23, suggesting little difference between the two types of forests.

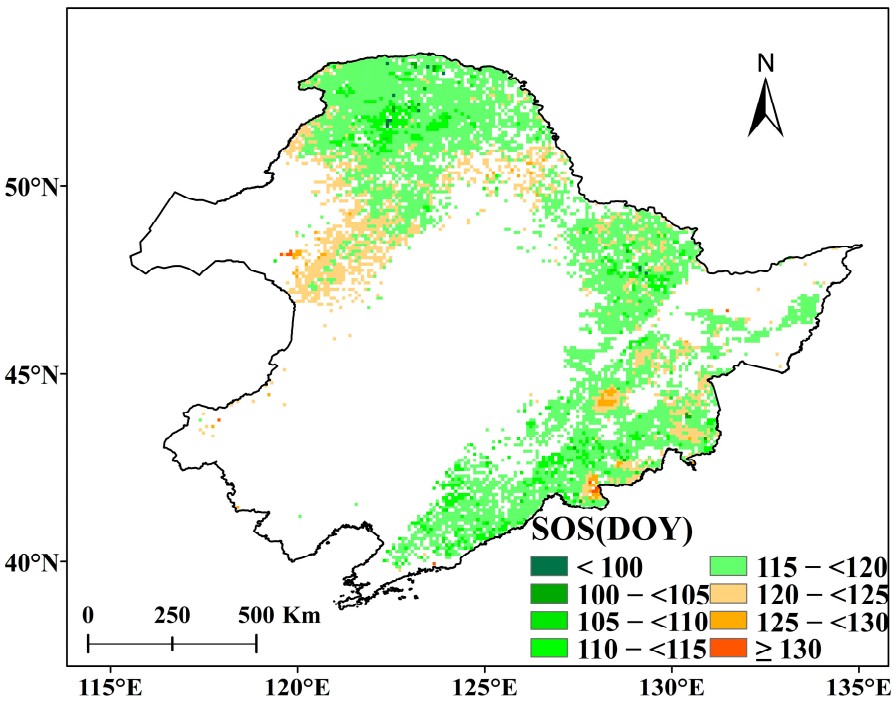

**Figure 2.** Spatial pattern of long-term average SOS of forests in Northeast China.

In Figure 3, we calculated the average SOS and its change during the analysis period. In the 32 years from 1982 to 2013, the spring phenology of the two main forests in Northeast China both showed an overall advancing trend, but only DNF showed a significant trend ($-0.184$ days per annum, $p < 0.05$), while the trend for DBF was not significant ($-0.093$ days/a, $p > 0.10$). However, the 32-year trends mask a huge inconsistency between the changes in SOS in DNF and DBF before and after 1998. In the earlier rapid warming period (1982–1998) prior to the short-term slowdown, the SOS of DNF rapidly advanced at the rate of 0.428 days/a ($p < 0.05$), while the trend of DBF was not significant ($-0.313$ days/a, $p > 0.10$). By contrast, during the short-term slowdown (1998–2013), it was the SOS of DBF that showed a significant trend—and a positive one, with a rate of 0.491 days/a ($p < 0.10$)—while the SOS of DNF showed an insignificant trend (0.169 days/a, $p > 0.10$).

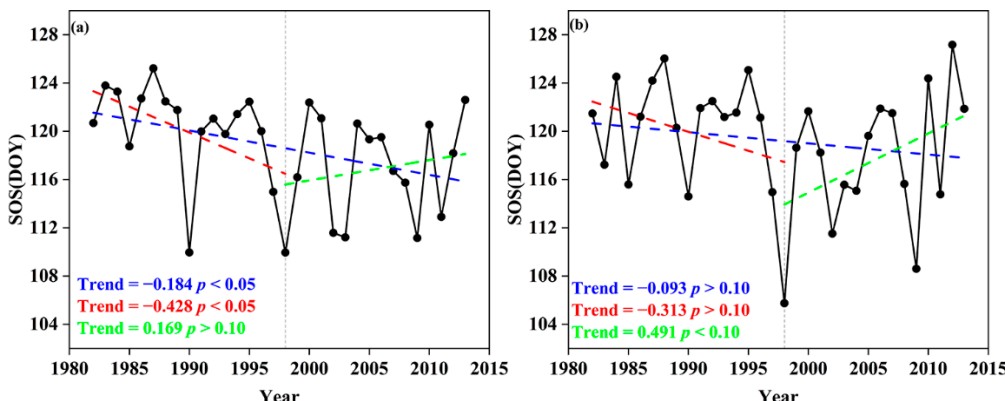

**Figure 3.** Trends of average SOS from 1982 to 2013: (**a**) deciduous needleleaf forests; (**b**) deciduous broadleaf forests. The solid black line represents the annual average SOS series from 1982 to 2013. The blue dashed line represents the SOS trend line from 1982 to 2013, the red dashed line represents the SOS trend line from 1982 to 1998, and the green dashed line represents the SOS trend line from 1998 to 2013.

From the earlier rapid warming period (1982–1998) to the short-term slowdown (1998–2013), the SOS of Northeast China forests appears to have transitioned from advancing to delaying but was driven by changes between the two forest types. Focusing only on statistically significant trends, DNF advanced rapidly before the short-term slowdown, then DBF delayed rapidly during the short-term slowdown.

Figure 4 presents, on a pixel-by-pixel basis, the spatial change trend of SOS of the two forest types in the above three periods. We observe that the trends display significant zonal characteristics.

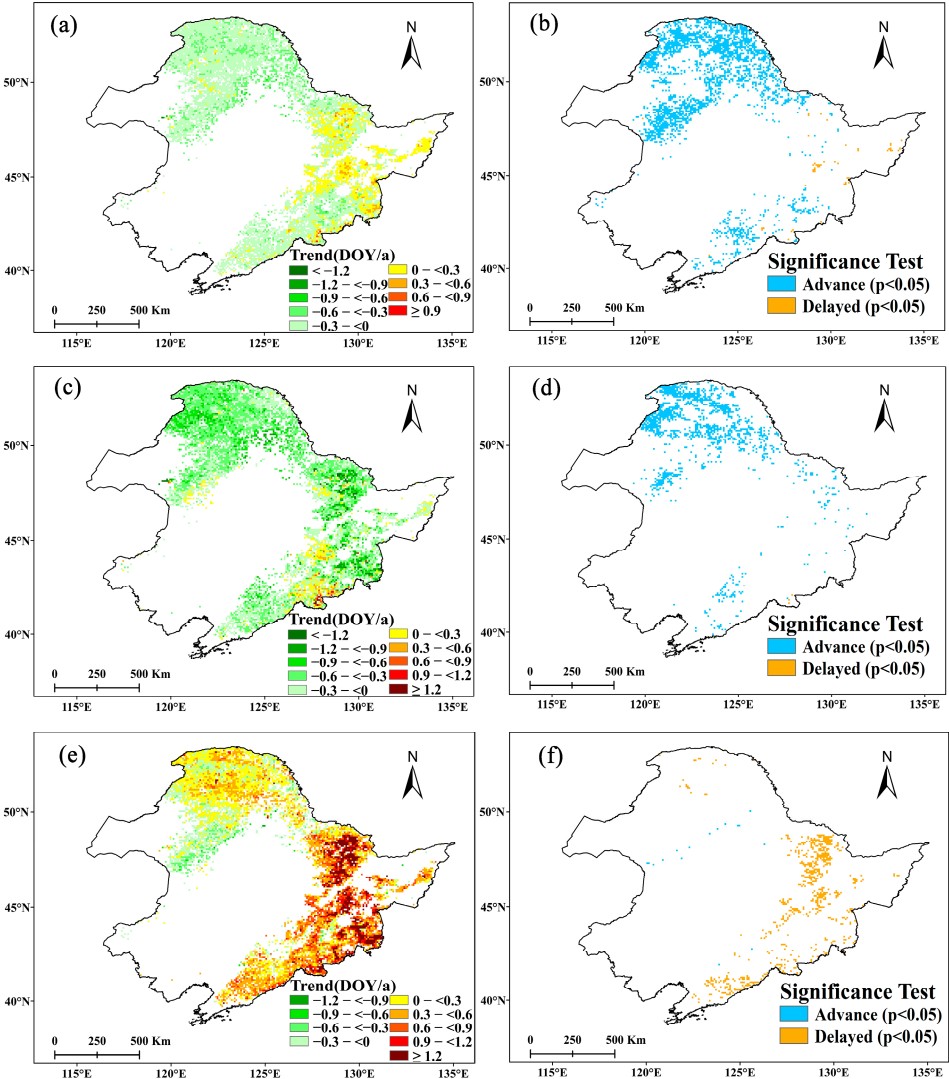

**Figure 4.** SOS trends (left) and significance test (right) in Northeast China forests, during the entire study period (1982–2013, (**a**,**b**)), the early rapid warming period (1982–1998, (**c**,**d**)) and the short-term slowdown (1998–2013, (**e**,**f**)).

Considering the entire study period of 1998–2013, the northeast forest SOS overall showed an advancing trend, but in the northern region more pixels passed the significance test of $p < 0.05$, while in the southern region fewer pixels passed the significance test. From 1982 to 1998, the northeast forests as a whole still showed an advancing trend and again more pixels in the northern forest region passed the significance test ($p < 0.05$), but in the southern region, only sporadic pixel changes were significant.

From 1998 to 2013, there was a significant difference from the previous period. The northeast forest SOS overall showed a delaying trend, but it was obvious that the trend was stronger in southern regions, such as the Changbai and Lesser Xing'an Mountains. The

significant delay ($p < 0.05$) was mainly concentrated in the southern part of the study area; in the north around the Great Xing'an Mountains, almost no pixels showed a significant delay.

The forests in the northern part of the study area are mainly composed of DNF, while those in the south are mainly DBF. Due to the obvious north–south characteristics of the spatial distribution of DNF and DBF, the obvious spatial differences in the three periods are strongly consistent with the results in Figure 3.

### 3.2. Relationships between SOS and Climatic Factors

We calculated the partial correlation coefficients of Tmax, Tmin, and precipitation (PCP) on the two forest types' SOSs in four seasons from the summer of the previous year to the relevant spring for 1982–2013, 1982–1998, and 1998–2013 (Figures 5 and 6). The influence of climate factors on forest SOS is quite different when analyzed seasonally.

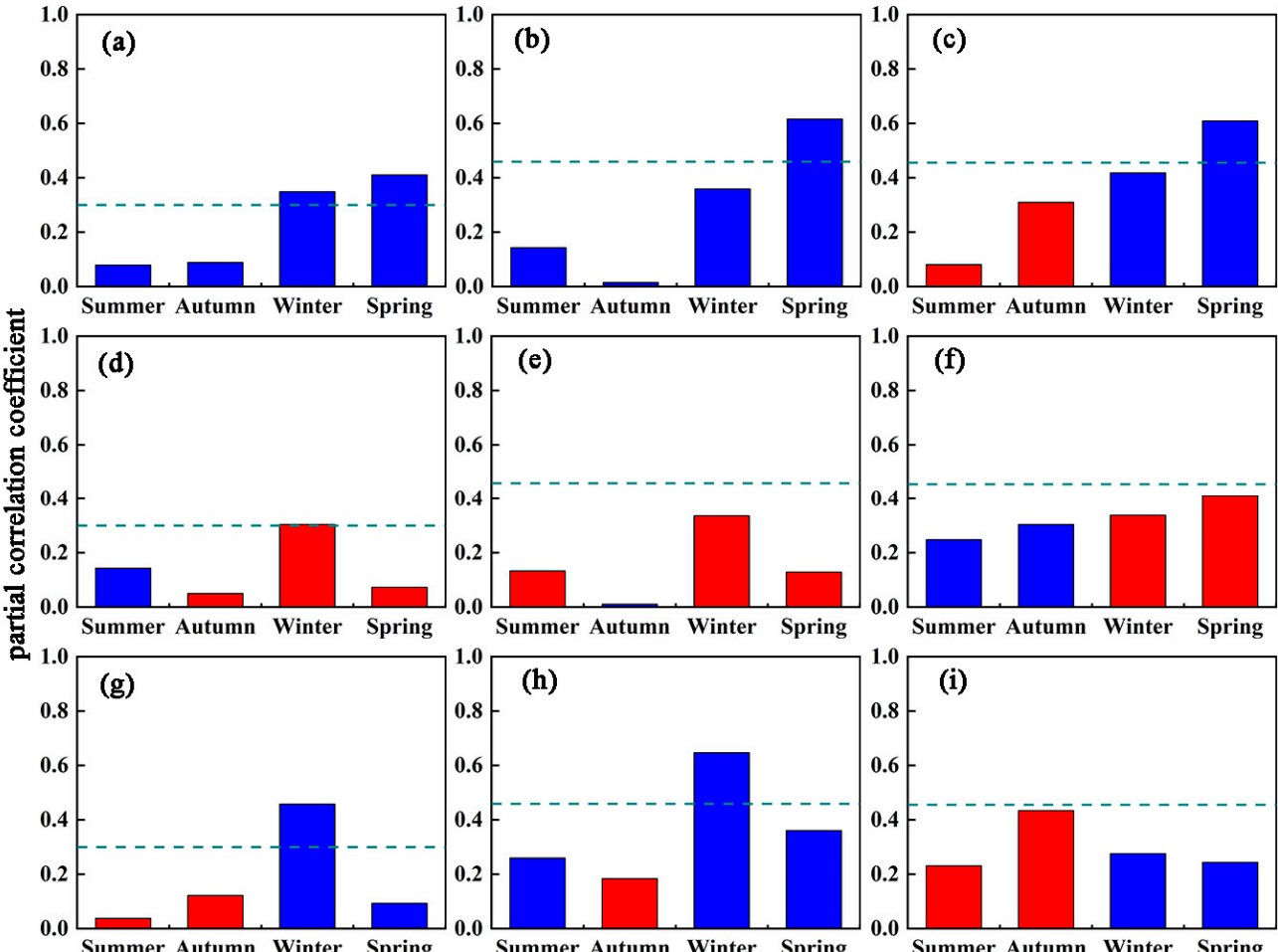

**Figure 5.** The response of deciduous needleleaf forest starting date of the growing season (DNF SOS) to changes in Tmax (**a–c**), Tmin (**d–f**), and precipitation (**g–i**) for the entire study period of 1982–2013 (graphs (**a,d,g**) at left), earlier rapid warming period to 1998 (graphs (**b,e,h**) in the middle), and short-term slowdown from 1998 (graphs (**c,f,i**) at right). Dashed lines indicate a significance level of 0.10; blue bars indicate negative correlations; red indicates positive correlations.

Tmax and Tmin have obvious asymmetric effects on SOS for the two forest types. The two forest types' SOSs are closely related to Tmax, primarily the spring Tmax, which has a significant negative effect (i.e., advancing SOS) for both forest types in all periods. The rise of Tmax in the spring can effectively promote the advancing forest SOS. Winter Tmax was also significantly negatively correlated with SOS for DNF when considering the entire study period, but not for either of the two subperiods.

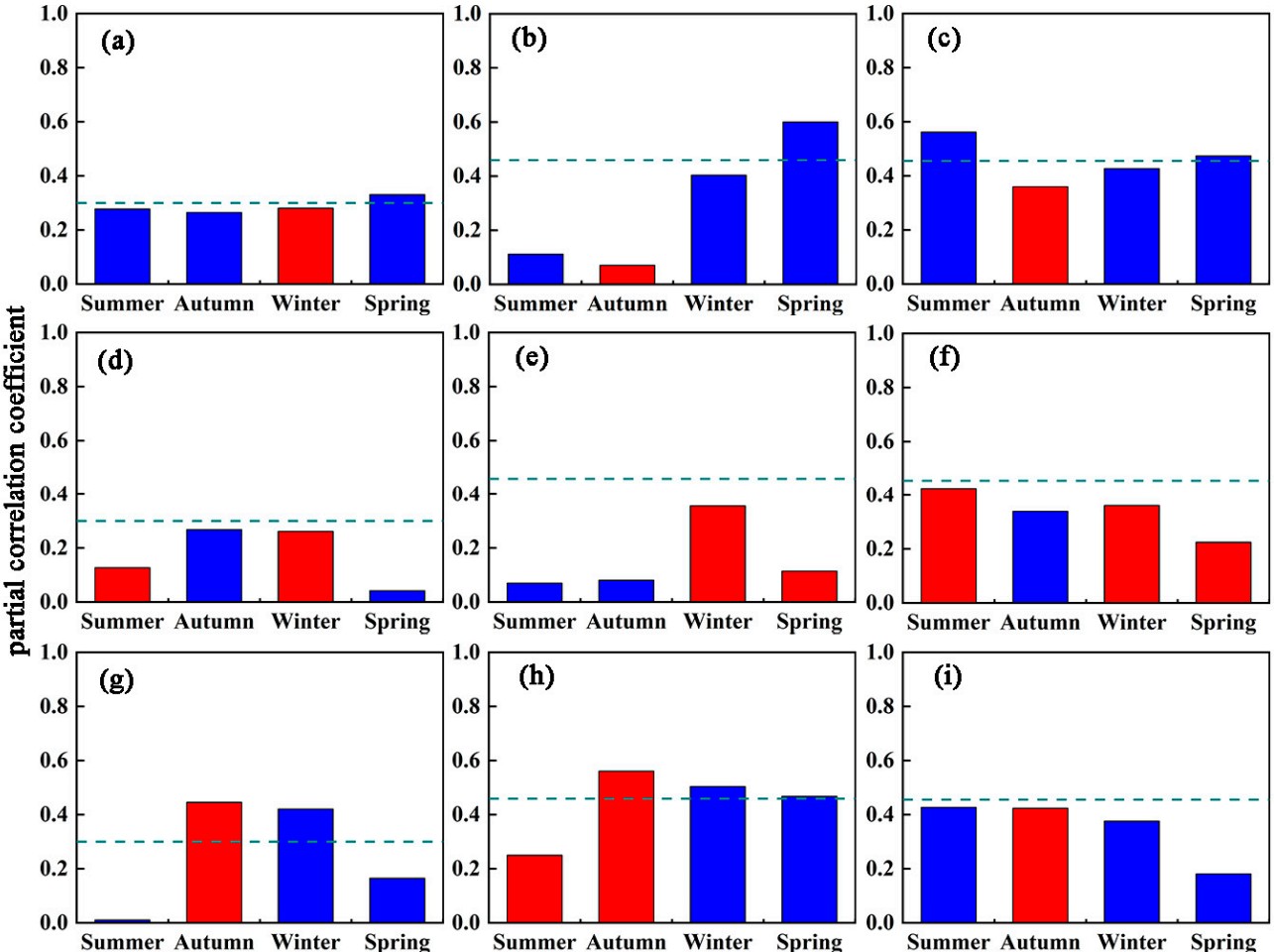

**Figure 6.** The response of deciduous broadleaf forest starting date of the growing season (DBF SOS) to changes in Tmax (**a**–**c**), Tmin (**d**–**f**) and precipitation (**g**–**i**) for the entire study period of 1982–2013 ((**a**,**d**,**g**) at left), earlier rapid warming period to 1998 ((**b**,**e**,**h**) in the middle), and short-term slowdown from 1998 ((**c**,**f**,**i**) at right). Dashed lines indicate a significance level of 0.10; blue bars indicate negative correlations; red indicates positive correlations.

Compared with Tmax, forest SOS appears less sensitive to Tmin. Only DNF SOS and winter Tmin showed a significant positive correlation between 1982 and 2013, opposite from the impact of Tmax. For the 1982–1998 and 1998–2013 subperiods, the partial correlation coefficients of SOS and Tmin in DNF were not significant, and for the DBF forest type, the SOS had no significant relationship with Tmin at any time.

Forest SOS is also closely related to precipitation. For DNF SOS, there was a significant negative correlation with winter PCP over the entire 1982–2013 study period and in the subperiod before the short-term slowdown, but no significant correlation with PCP during the short-term slowdown from 1998. The relationship between DBF SOS and PCP is relatively complex: from 1982 to 2013, DBF SOS is mainly positively correlated with autumn PCP ($p < 0.05$), and significantly negatively correlated with winter PCP ($p < 0.05$), while from 1982 to 1998, DBF SOS is significantly positively correlated with autumn PCP ($p < 0.05$), and significantly negatively correlated with winter PCP and spring PCP ($p < 0.10$). During the short-term slowdown, similar to DNF, DBF SOS has no obvious relationship with PCP.

The distributions of the two forest types differ spatially and are related to general climate conditions. The response of forest SOS to climate change in different regions also varies. We find that the SOS trends for DNF are only significantly related to climate factors in winter and spring, while for DBF, Tmax in summer and PCP in autumn also have an

impact. This suggests that DBF SOS is more affected by the legacy of the previous growing seasons relative to DNF.

Before the short-term slowdown, the SOSs of the two forest types were both sensitive to winter PCP and summer Tmax. When entering the short-term slowdown, as temperatures rose more slowly or even decreased, the SOS of both DNF and DBF maintained a high sensitivity to spring temperatures, but the sensitivity to PCP decreased to a nonsignificant level. Before the short-term slowdown, the SOS of DNF was significantly negatively correlated with spring precipitation, while the SOS of DBF was not only controlled by winter PCP, but also significantly positively correlated with autumn PCP of the previous year. Entering the period of short-term slowdown, the negative correlation effect of summer Tmax on SOS in the previous year was significantly enhanced for DBF, but this was not reflected in DNF.

By studying the change trends of climate factors (Tables 1 and 2), we found that before 1998, among the climate factors significantly related to DNF, winter PCP showed a significant increasing trend ($p < 0.05$). This was the main climate factor leading to the significant advance of DNF SOS. During the earlier rapid warming subperiod, the climate factors that have a significant relationship with DBF did not change significantly, so it is no surprise to find no significant change in DBF prior to 1998.

**Table 1.** Trend of climate factors in DNF.

|  | Year | Summer | Autumn | Winter | Spring |
|---|---|---|---|---|---|
| Tmax | 1982–2013 | <u>0.067</u> | 0.043 | −0.050 | 0.000 |
|  | 1982–1998 | 0.079 | **0.098** | 0.069 | 0.087 |
|  | 1998–2013 | 0.004 | 0.070 | **−0.191** | −0.087 |
| Tmin | 1982–2013 | <u>0.031</u> | 0.012 | −0.045 | 0.020 |
|  | 1982–1998 | 0.028 | 0.050 | 0.070 | 0.097 |
|  | 1998–2013 | 0.026 | 0.038 | −0.164 | −0.071 |
| PCP | 1982–2013 | <u>−2.228</u> | −0.226 | <u>0.156</u> | 0.275 |
|  | 1982–1998 | −2.817 | 0.388 | <u>0.302</u> | −0.114 |
|  | 1998–2013 | 1.921 | −0.475 | 0.219 | 1.074 |

Values in boldface indicate $p < 0.10$, underlined values indicate $p < 0.05$, double underlined values indicate $p < 0.01$.

**Table 2.** Trend of climate factors in DBF.

|  | Year | Summer | Autumn | Winter | Spring |
|---|---|---|---|---|---|
| Tmax | 1982–2013 | **0.036** | <u>0.039</u> | −0.030 | −0.010 |
|  | 1982–1998 | 0.041 | 0.061 | **0.103** | 0.077 |
|  | 1998–2013 | −0.021 | 0.039 | −0.171 | **−0.142** |
| Tmin | 1982–2013 | <u>0.030</u> | <u>0.035</u> | −0.013 | 0.025 |
|  | 1982–1998 | 0.034 | 0.060 | <u>0.107</u> | **0.083** |
|  | 1998–2013 | 0.002 | 0.046 | −0.135 | −0.082 |
| PCP | 1982–2013 | <u>−1.958</u> | −0.338 | <u>0.231</u> | **0.761** |
|  | 1982–1998 | −1.294 | 0.859 | 0.011 | −0.315 |
|  | 1998–2013 | 1.022 | 1.605 | 0.431 | 1.489 |

Values in boldface indicate $p < 0.10$, underlined values indicate $p < 0.05$, double underlined values indicate $p < 0.01$.

In the short-term slowdown starting in 1998, only Tmax in spring is closely related to DNF SOS, but its downward trend is not significant ($p > 0.10$), so the delay of DNF SOS is not obvious ($p > 0.10$). The relationship between DBF and Tmax in the spring and the previous summer is significant: as spring Tmax decreases significantly ($p < 0.10$) it can be regarded as a main contributor leading to a significant delay in SOS in the broad-leaved forest ($p < 0.10$).

## 4. Discussion

While we find little difference between the spring phenology of the two Northeast China forest types examined here, the SOS for both forest types is trending earlier —for the study period of 1982 to 2018, the average SOS for deciduous coniferous forest is DOY 118.69, compared less than one full day earlier than that of the deciduous broad-leaved forest, DOY 119.23. These results are similar to those found in prior studies in Northeast China [10,20] and consistent with the widely reported advance of SOS in the high latitudes of the Northern Hemisphere since the 1980s [11,33,37].

Interestingly, we find that the trends in both forest types, DNF and DBF, reversed around 1998. The rate of change of SOS in DNF changed from significantly advancing ($-0.428$ days/a, $p < 0.05$) in 1982–1998 to insignificantly delaying (0.169 days/a, $p > 0.10$) in 1998–2013, while DBF changed from an insignificant advance ($-0.313$ days/a, $p > 0.10$) to a significant delay (0.491 days/a, $p < 0.10$). Wang et al. (2019) [20], in studying the changes of overall vegetation SOS in the Northern Hemisphere from 1982 to 2014, found that the rate of advance of SOS in the Northern Hemisphere decreased after 1998, reporting a small and weakly significant delaying trend ($-0.020$ days/a, $p = 0.967$). Our study confirmed that there were spatial differences in different regions of the northern hemisphere, and the delay rate of Chinese northeast forests' SOSs during the short-term slowdown was stronger than that of the Northern Hemisphere as a whole.

Shen et al. (2018) [13] also cautioned that trend calculations are sensitive to the selection of study periods or subperiods. In our study, we determined that extending the analysis window for the short-term slowdown subperiod by one year (to 1998–2014 instead of 1998–2013) resulted in a smaller rate of delay in the DBF SOS (0.170 days/a instead of 0.491 days/a) and reversed the trend for DNF SOS (to $-0.112$ days/a). We focus here on the shorter subperiod, including the 2013 growing season, but not 2014, because of the hypothesized effect of the prior year's climate conditions on the SOS, given that the short-term slowdown is often considered to be 1998–2012 [16,38].

Large-scale studies tend to mask the characteristics of differences at smaller scales. Deng et al. (2019) [5] found that from 2001 to 2017, DNF in high latitudes in the Northern Hemisphere experienced a gradual delay in SOS, similar to our results in Figure 6e. Interestingly, we found that the SOS delay in DBF, found at lower latitudes than DNF, was stronger than DNF in the same period, indicating that forest types will respond differently to climate change.

For the entire research period, spring Tmax is the key factor affecting the SOS of these forest types, both before and during the short-term slowdown. Temperature is often considered to play an important role in SOS in high latitudes [39]. Tmax often drives heat accumulation before the start of the growing season, promoting germination [11,36]. We find that during the short-term slowdown, the decline of the high spring temperatures is the main factor leading to the delay in SOS in both forest types. The decreasing trend of Tmax in spring in DNF ($-0.087\,°C/a$, $p > 0.10$) is significantly smaller than that in DBF ($-0.142\,°C/a$, $p < 0.10$), so the SOS delay rate (0.169 days/a, $p > 0.10$) in DNF is also smaller than that in DBF (0.491 days/a, $p < 0.10$).

The asymmetric effect of daytime and nighttime temperatures on vegetation SOS has also been confirmed in the northeast China forest region. Although we find that the influence of Tmin on SOS is weak, for both DNF and DBF there is a positive correlation between forest SOS and Tmin in winter and even spring. That is, the higher the Tmin, the later the SOS. This is consistent with the report of Meng et al. (2020) [40] that Tmin and Tmax play opposite roles in affecting forest SOS. This may relate to vegetation sensitivity to winter chilling—that is, plants are adapted to meet a certain cold demand in winter before they can be green normally in spring, so higher minimum temperatures may delay SOS by delaying the satisfaction of the cold demand [29,41]. Winter Tmin in Northeast China is very low, so Tmin change does not constitute a limiting effect on changing forest SOS. Increases in spring Tmin, though, may also promote respiration, thus inhibiting the greening process of vegetation: excessive respiration will consume a large amount of the

organic material accumulated by photosynthesis, thereby slowing down the process of plant greening, which will in turn lead to a later SOS as detected by NDVI [42,43].

Before the short-term slowdown, the SOS of both forest types was also closely related to winter PCP. Winter precipitation in Northeast China is often in the form of snow, providing an important source of soil moisture in the next growing season [44,45]. More PCP in winter may provide sufficient water during the next snowmelt to promote germination of vegetation. In winter, the photosynthesis of deciduous forests often stagnates, and the increase in PCP can inhibit respiration by reducing the temperature, causing vegetation to retain more accumulated organic matter [46,47].

Autumn PCP showed a significant positive correlation with DBF SOS across the entire study period (1982–2013) and in the rapid warming subperiod (1982–1998). Vigorous precipitation in autumn, typically the end of the growing season, can delay the seasonal pause in vegetation growth by alleviating soil water stress [48]. However, this extended growth period may leave DBF susceptible to injury from freezing during the winter, and thus affect the next year's growing season.

Under more arid conditions, vegetation tends to become more sensitive to water [13,14,49]. Over our entire study period, the winter PCP in both DNF and DBF regions showed a significant increasing trend ($p < 0.05$), contributing to the advance in SOS. However, during the short-term slowdown subperiod, sensitivity to water supply appears to be connected with thermal conditions, weakening the effect. Within a certain range, higher temperature will often increase the water demand of plants [13,49]; decreasing temperatures in the short-term slowdown may have also reduced plants' demand for water and removed the restrictive effect of water conditions. Some studies have also pointed out that increasing global atmospheric $CO_2$ concentrations have led to the reduction of stomatal conductance of vegetation leaves: plants can exchange less water for the same amount of carbon, thus significantly reducing the water demand of vegetation growth [50]. All in all, the winter PCP no longer had a significant impact on Northeast China forest SOS during the short-term slowdown.

Vegetation growth is not only affected by the climatic conditions of the current year, but also by the growth conditions of the previous growing season [2,32]. In this study, we found that the DNF SOS is closely related only to the climate factors of the winter and spring, and weakly related to the climate factors of the previous summer and autumn. This indicates that the SOS of the DNF forest type is less affected by its growth in the previous year. In comparison, DBF is more closely related to some climate factors in the previous summer and autumn, indicating that the previous year's growing season has a stronger impact on the SOS of DBF.

With the end of the short-term slowdown, the global warming trend is expected to continue to accelerate, affecting forest temperature and precipitation conditions worldwide. A study of the broadleaf forests of Yunnan and Guizhou in southwest China [1] pointed out that the degree of drought in summer will directly affect the end date of the growing season, with likely consequences for the subsequent year's SOS. Under conditions of further warming, the SOS of DNF and DBF in Northeast China may again see some of the characteristics we observed before the short-term slowdown—that is, the SOS of DNF will advance faster, while the SOS of DBF will advance slightly slower. The IPCC [51] has pointed out that, while global warming is a well-documented long-term trend, it is neither spatially nor temporally uniform. In addition to greenhouse gases, changes in atmospheric circulation, the water cycle, and the El Niño Phenomenon may produce short-term climate fluctuations. Our research can be helpful to modeling the impact of these variations on vegetation under global warming scenarios.

## 5. Conclusions

From 1982 to 2013, the SOS of the two main forest types in northeast China showed an advancing trend. However, considering the subperiods before and after 1998, the trends diverge from advancing to delaying. There are differences between different forest types:

in the earlier subperiod of rapid warming, the DNF SOS advanced quickly while the DBF SOS advanced slowly if at all; during the subsequent short-term slowdown, the DBF SOS was delayed significantly while the DNF SOS experienced no significant delay. Tmax and Tmin have different effects: SOS has a close negative correlation with Tmax in the spring but is less affected by Tmin. The role of PCP cannot be ignored. From 1982 to 1998, the advance of SOS in DNF was mainly caused by winter PCP, and the slow rise of winter PCP and spring Tmax led to the slow advance of SOS in DBF during this period.

During the short-term slowdown period, spring Tmax decreased; the significant correlations of DNF and DBF SOS with precipitation disappeared, but they were still highly sensitive to Tmax in spring. During this period, the SOS of DNF showed a slow delay, but the SOS of DBF delayed faster due to the rapid decline of Tmax in spring.

Regional differences in the distribution of the two forest types result in obvious spatial variations in the changing trends of SOS in the northeast forests before and during the short-term slowdown. DNF SOS is less affected by the climatic conditions of the previous growing season, while DBF is more affected. With the resumption of global warming trends following the short-term slowdown, we may expect that the SOS of DNF may again advance faster than that of DBF.

**Author Contributions:** Conceptualization, F.Z. and B.L.; methodology, F.Z. and X.S.; software, F.Z., Y.S. and W.Z.; validation, B.L.; formal analysis, F.Z.; writing—original draft preparation, F.Z.; writing—review, M.H. and B.L. All authors have read and agreed to the published version of the manuscript.

**Funding:** This research was funded by the National Natural Science Foundation of China (41877416) and Youth Innovation Promotion Association, Chinese Academy of Sciences (2019235).

**Acknowledgments:** We gratefully acknowledge the National Natural Science Foundation of China and Youth Innovation Promotion Association, Chinese Academy of Sciences for funding this work.

**Conflicts of Interest:** The authors declare no conflict of interest.

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
