# Peer review of "Changing Spring Phenology of Northeast China Forests during Rapid Warming and Short-Term Slowdown Periods"

_forests, doi:10.3390/f13122173_

Round 1

Reviewer 1 Report

A well written manuscript and nice study focused on disentangling long term patterns and drivers in phenology across broadleaf needle and deciduous forests in China. The results match general patterns of earlier leaf expansion in the northern hemisphere, yet provide nuanced insight into differential responses between forest types and the potential drivers. 

Author Response

Response to Reviewer 1 Comments:

Point 1:A well written manuscript and nice study focused on disentangling long term patterns and drivers in phenology across broadleaf needle and deciduous forests in China. The results match general patterns of earlier leaf expansion in the northern hemisphere, yet provide nuanced insight into differential responses between forest types and the potential drivers.

Response1: Thank you for your favorable comments!

Reviewer 2 Report

This is a relatively well written and presented paper. My major concern is with the use of the concept of the warming hiatus, which has now been discredited (see inter alia the references below). I suspect that the use of this artificial disconnect in a warming trend accounts for some of the confusing statistical results found (as noted in comments below). I would advise that, at the very least, the analyses be repeated without the breakdown of the data set into two periods, especially given that the official (IPCC) view is that the hiatus never happened. For this reason I have recommended a major revision

Some other issues:

Line 40: ‘recent decades’ is vague, give the dates.

Line 43: ‘hot topic’ is poor expression.

Line 46: the definition of SOS is different in the Abstract.

Line 53: here, and elsewhere, ‘northern hemisphere’ should have capita initial letters (Northern Hemisphere). This is inconsistent throughout the MS.

Linde 55 and elsewhere: ‘nighttime’ is conventionally written as ‘night time’.

Line 61: delete ‘is’.

Line 66: I cannot easily check these citations, but it is certain that these authors did not define the term ‘warming hiatus’.

Line 84: delete comma.

Line 123: space required between number and unit.

Line 170: the legend in the figure does not make sense. Some numbers can appear in more than one category.

Line 188: what is the solid black line.

Line 212: The figure components are far too small to read clearly (I need a magnifying glass). It looks as if there is a similar issue with the legend to that noted above.

Line 221: I don’t understand this figure. The legend talks about three columns, but each graph has four. Acronyms need defining in the legend.

Line 237: same issue with the figure as noted above.                 

Line 251: delete ‘are’ before ‘differ’.

Lines 279 and 281; the alignment of numbers In some of the columns of the tables needs attention.

Lines 282-289 (and elsewhere): these statistics/findings are confusing and may result from an artificial breakdown of the data set for analysis (see comments regarding the warming hiatus above). I suspect this also accounts for the weak relationships described during the “hiatus” period. Because of this the reliability and meaning of the findings cannot be relied upon, at least not without careful re-examination of the data. It may be worth considering reinterpreting the findings in the light of the question ‘if the hiatus is real, did it effect the phenology’, with the answer probably being ‘not strongly, if at all’! At the very least, a comparative analysis of the whole data set as one run needs to be performed to compare the findings, and to see the impact of analysing on the basis of the “hiatus”.

Line 331: delete ‘be’.

Line 372: move citation to reference [1] to after ‘China’.

Cahill, Niamh; Rahmstorf, Stefan; Parnell, Andrew C (1 August 2015). "Change points of global temperature"Environmental Research Letters10 (8): 084002. Bibcode:2015ERL....10h4002Cdoi:10.1088/1748-9326/10/8/084002.

Hansen, James; Sato, Makiko; Ruedy, Reto; Schmidt, Gavin A.; Lob, Ken (19 January 2016). "Global Temperature in 2015" (PDF).

Lewandowsky,  Stephan; James S. Risbey; Naomi Oreskes (2015)"On the definition and identifiability of the alleged "hiatus" in global warming: Scientific Reports". Nature. doi:10.1038/srep16784.

Wendel, JoAnna (2015). "Global warming "hiatus" never happened, study says". Eos. 96doi:10.1029/2015EO031147.

 Wolf, Eric; et al. (27 February 2014). Climate Change, Evidence & Causes (PDF). Royal Society (UK) and National Academy of Science (USA).

 "Study confirms steady warming of oceans for past 75 years". PhysOrg. 4 January 2017.

 "E-Library: WMO Statement on the status of the global climate in 2013"World Meteorological Organisation. 24 March 2014

Author Response

We thank you for the time and effort that you have put into reviewing the previous version of the manuscript. We carefully modified the paper in response to the comments and suggestions. Please see the attachment.

Reviewer 3 Report

Your manuscript presents an study focused on forests SOS dates, which differs from articles that deal with similar topics (as far as I am aware). It was well written and I have few questions and remarks.

- Are these four seasons? ..."with four distinct seasons; long, cold winters; short, warm summers". Please clarify.

- The ESA land cover dataset present any classification accuracy estimate?

- In figures 5 and 6 please complete the axis (partial correlation coefficient?)

- I wonder if Tables 1 and 2 could be improved with information on increasing or decreasing trend for the significant factors?

- Please explain that "Increases in spring Tmin.....and inhibit the greening process of vegetation".

- I always consider wise, in works using datasets based on several sources, which certainly are not 100% controlled, to write about possibilities of uncertainty, even for significant statistics.

- You could go further perhaps when discussing results, trying to imagine the effects of delaying/advancing SOS for forests in face of global warming, maybe considering IPCC scenarios. It is still vague to imagine the consequences of your results for vegetation, and perhaps mankind.

Author Response

(The authors gave the same response as above.)

Reviewer 4 Report

This study presents the changes of spring phenology in Northeastern China and obtained some useful results. It is a meaningful study.

There are also some questions that must be revised:

1, How do you define the “warming hiatus periods”? In my opinion, it must a long time and I don’t think from 1998 to 2013 is warming hiatus periods.

2, Why you use NDVI to study vegetation changes in forest? Do you know the “saturation”? and the temporal range of GIMMS NDVI is 1982-2015, why you used 1982-2013? The spatial resolution is 8km, I don’t think it satisfy the study project. Why not select MODIS EVI?

3, How do you select DNF and DBF of the study region?

4, Do you smooth the time series data? Reference: “Vegetation Phenology in Permafrost Regions of Northeastern China Based on MODIS and Solar-induced Chlorophyll Fluorescence”

5, Why you select the threshold value of 0.2 for SOS? Is that fit the study region?

6, In section 3.2 why you calculated the partial correlation coefficient for four seasons? Why not just spring?

7, Tile of tables and figures are not representing the meaning of that.

Author Response

(The authors gave the same response as above.)

Round 2

Reviewer 2 Report

My thanks to the authors for addressing my general questions: it is much easier to obtain a sense of the science now.

However, I still have some concerns.

The authors are persisting on the use of the term ‘hiatus’ to describe the slowing down (not the stopping!) of the warming trend from 1998 to 2013. As I noted, with supporting references, this term is now obsolete (see also a recent joint (2020) report by the Royal Society and the National Academy of Scienceshttps://royalsociety.org/~/media/Royal_Society_Content/policy/projects/climate-evidence-causes/climate-change-evidence-causes.pdf. It may b that Chinese scientists are still using the term, but is, as noted previously, now recognised as being an incorrect description of what happened. I cannot recommend publication of a paper using incorrect terminology.

There is also a major flaw in the data analysis. On the basis of the ‘hiatus’ existing, the authors divide their data analysis into three blocks; the whole period 1982-2013, pre-‘hiatus’ 1982-1998 and during the ‘hiatus’ 1998-2013. This means that the data for 1998 were analysed twice, as they appear in the second and third analysis periods. This is confusing at best, and misleading/wrong at worst, adding uncertainty and unreliability to the conclusions presented.

Also, the intervals noted for the  day of year (DOY) plots in Figures 2 and 4 are incorrect: the same values could appear in more than one category. I flagged this previously and was told that the issue had been corrected, but the error is still present in the version of the MS I am seeing.

Finally, line 145: the parameters are not defined, the reader has no idea what these are.

Tangentially, I don’t know why the text in lines 131-141 is in italics. And ‘obtained’ should be deleted from line 105.

Author Response

Thank you very much for your careful review. Your comments have greatly improved the quality of our papers. At the same time, in the process of communication with you, they have also improved our vision, allowing us to have a deeper understanding of the latest science. We have revised the paper  based on your suggestions and explained some problems. I am very much looking forward to this revision and reply meeting your requirements, as well as the publication of this paper. Finally, I would like to express our sincere thanks to you again.

Point 1: The authors are persisting on the use of the term ‘hiatus’ to describe the slowing down (not the stopping!) of the warming trend from 1998 to 2013. As I noted, with supporting references, this term is now obsolete (see also a recent joint (2020) report by the Royal Society and the National Academy of Scienceshttps://royalsociety.org/~/media/Royal_Society_Content/policy/projects/climate-evidence-causes/climate-change-evidence-causes.pdf. It may b that Chinese scientists are still using the term, but is, as noted previously, now recognised as being an incorrect description of what happened. I cannot recommend publication of a paper using incorrect terminology.

Response 1: Thank you for your suggestion, which has let us know the latest progress in the field of climate change, as well as the recent progress made in the controversy around the so-called warming or stagnation, pause, or hiatus in the early years of the 21st century. According to your suggestion, we replaced the term “warning hiatus” with references to a “short-term slowdown” that is more in line with contemporary science. Our research is more concerned with the fact that, in our study area, there are obvious differences in the seasonal changes in temperature during the two subperiods, giving us a chance to see the effects of different warming profiles on seasonal .

Point 2: There is also a major flaw in the data analysis. On the basis of the ‘hiatus’ existing, the authors divide their data analysis into three blocks; the whole period 1982-2013, pre-‘hiatus’ 1982-1998 and during the ‘hiatus’ 1998-2013. This means that the data for 1998 were analysed twice, as they appear in the second and third analysis periods. This is confusing at best, and misleading/wrong at worst, adding uncertainty and unreliability to the conclusions presented.

Response 2: Thank you for your question. We have also thought a lot about the breakpoint question you raised in the process of writing the paper. We have referred to a large number of recent studies of vegetation response to climate. We found that in the relevant papers on this topic, the analysis methods of breakpoints included in the previous two periods were most used. We have given the relevant literature we have reviewed (below). In addition, due to the strong relationship between sos and seasonal climate change (below), the intersection of the two periods in 1998 can better show the difference between the two periods of seasonal climate change, which can be said to expand the difference signal.

Guo, Z.; Lou, W.; Sun, C.; He, B. Trend Changes of the Vegetation Activity in Northeastern East Asia and the Connections with Extreme Climate Indices. Remote Sens. 2022, 14, 3151.. Remote Sens. 2022, 14, 3151.

Li, M.; Yao, J.; Guan, J.; Zheng, J. Vegetation Browning Trends in Spring and Autumn over Xinjiang, China, during the Warming Hiatus. Remote Sens. 2022, 14, 1298.

Li, Y.; Zhang, Y.; Gu, F.; Liu, S. Discrepancies in vegetation phenology trends and shift patterns in different climatic zones in middle and eastern Eurasia between 1982 and 2015. Ecol Evol. 2019, 9, 8664-8675.

Walther, G.R.; Post, E.; Convey, P.; Menzel, A.; Parmesan, C.; Beebee, T.J.C.; Fromentin, J.M.; Hoegh-Guldberg, O.; Bairlein, F. Ecological Responses to Recent Climate Change. Nature 2002, 416, 389–395.

Wang, X.; Xiao, J.; Li, X.; Cheng, G.; Ma, M.; Zhu, G.; Arain, M.A.; Black, T.A.; Jassal, R. S. No trends in spring and autumn phenology during the global warming hiatus. Nat Commun. 2019, 10, 1-10.

Point 3: Also, the intervals noted for the day of year (DOY) plots in Figures 2 and 4 are incorrect: the same values could appear in more than one category. I flagged this previously and was told that the issue had been corrected, but the error is still present in the version of the MS I am seeing.

Response 3: Thank you for your suggestion. In the first revision, we didn't fully understand your meaning. We apologize here. In this modification, we adjusted the legend according to your requirements to clarify that the categories go up to but do not include the upper value (e.g., 100 - <105).

Point 4: Finally, line 145: the parameters are not defined, the reader has no idea what these are.

Response 4: Thank you for your suggestion. The description in our last manuscript is really brief. We have added information about relevant parameters according to your requirements, which will make people more clear about this formula.

Point 5: Tangentially, I don’t know why the text in lines 131-141 is in italics. And ‘obtained’ should be deleted from line 105.

Response 5: I'm really sorry on the format problems in our adjustment. We have revised the manuscript according to your requirements. Thank you for your meticulous inspection.

Reviewer 4 Report

For the questions 2, 3 and 5, I can't agree with you.

Question 2, No matter the longer time scale. Your study region is not large enough and may be 1 km is more suitable. In forest, more studies used EVI but not NDVI.

Question 3, I don't think the spatial resolution of 300m is higher enough to distinguish DNF and DBF.

Question 5, For different regions, the threshold values should be different.

Author Response

Thank you very much for your careful review. Your comments have greatly improved the quality of our papers. At the same time, in the process of communication with you, they have also improved our vision, allowing us to have a deeper understanding of the latest science. We explained again in combination with your suggestions. This kind of communication is a great progress for us. I am very much looking forward to this reply meeting your requirements, as well as the publication of this paper. Finally, I would like to express our sincere thanks to you again.

Point 1: Question 2, No matter the longer time scale. Your study region is not large enough and may be 1 km is more suitable. In forest, more studies used EVI but not NDVI.

Response 1: Thank you for your question. On a global scale, our research coverage is relatively small, but on a regional scale, the total area of our research area exceeds 400,000 square kilometers. There have been many papers on similar scales using this set of data sources (below). In addition, the vegetation index data within 1km provided by MODIS and SPOT first appeared in 1998. Because these high-precision vegetation index data appeared too late, the research time range based on the vegetation index within 1km was after 1998 (below), which really cannot support the aim of research. Higher resolution EVI data would be useful for future studies. We thank you for your suggestions.

Kong, D.; Miao, C.; Wu, J.; Zheng, H.; Wu, S. Time lag of vegetation growth on the Loess Plateau in response to climate factors, Estimation, distribution, and influence. Sci. Total Environ. 2020, 744, 140726.

Shen, X.; Liu, B.; Xue, Z.; Jiang, M.; Lu, X.; Zhang, Q. Spatiotemporal variation in vegetation spring phenology and its re-sponse to climate change in freshwater marshes of Northeast China. Sci Total Environ. 2019, 666, 1169-1177.

Su, M.; Huang, X.; Xu, Z.; Zhu, W.; Lin, Z. A Decrease in the Daily Maximum Temperature during Global Warming Hiatus Causes a Delay in Spring Phenology in the China-DPRK-Russia Cross-Border Area. Remote Sens. 2022, 14, 1462.

Wang, Y.; Shen, X.; Jiang, M.; Lu, X. Vegetation Change and Its Response to Climate Change between 2000 and 2016 in Marshes of the Songnen Plain, Northeast China. Sustainability 2020, 12, 3569.

Wen, Z.; Wu, S.; Chen, J.; Lü, M. NDVI indicated long-term interannual changes in vegetation activities and their responses to climatic and anthropogenic factors in the Three Gorges Reservoir Region, China. Sci. Total Environ. 2017, 574, 947-959.

Yang, J.; Zhang, H.; Yang, W. Coupling Effects of Precipitation and Vegetation on Sediment Yield from the Perspective of Spatiotemporal Heterogeneity across the Qingshui River Basin of the Upper Yellow River, China. Forests 2022, 13, 396.

Point 2: Question 3, I don't think the spatial resolution of 300m is higher enough to distinguish DNF and DBF.

Response 2: Thank you for your question. The ESA land cover data we used is one of the most authoritative land classification data at present and has been adopted by many papers (below). Using this set of data, we can try our best to reduce the error of research results caused by vegetation cover change. The most important thing is that the intersection degree between DNF and DBF is very small, and the difference is obvious. There are only some intersections in high altitude areas, which can basically represent two kinds of climatic zones. We can use them to better present relevant research results. Thanks again for your question.

The ESA climate change initiative: Satellite data records for essential climate variables. Bull. Am. Meteorol. Soc. 2013, 94, 1541–1552.

Driving Forces of the Changes in Vegetation Phenology in the Qinghai–Tibet Plateau. Remote Sens. 2021, 13, 4952.

Point 3: Question 5, For different regions, the threshold values should be different.

Response 3: Thank you for your question. The selection of threshold is very important for remote sensing phenology. When Jönsson and Eklundh proposed the dynamic threshold method, they suggested that the beginning and end of the growing season should be about 20% of the annual amplitude of NDVI (below). This is well applied in high latitude and high altitude areas (below). At the same time, we used another commonly used threshold of 50% to test, and found that there was a certain difference in the absolute value of the date between the two, but in terms of long-term change trend, the phenomenon was basically similar, which is very important for our research, because we pay more attention to the change trend of SOS. To sum up, and in combination with our long-term local experience, we chose a more suitable 20% as our threshold. Thanks again for your question.

Hufkens, K.; Friedl, M.; Sonnentag, O.; Braswell, B.H.; Milliman, T.; Richardson, A.D. Linking near-surface and satellite remote sensing measurements of deciduous broadleaf forest phenology. Remote Sens. Environ. 2012, 117, 307–321.

Jonsson, P.; Eklundh, L. Seasonality extraction by function fitting to time-series of satellite sensor data. IEEE Trans. Geosci. Remote Sens. 2002, 40, 1824-1832

Liu, X.; Chen, Y.; Li, Z.; Li, Y.; Zhang, Q.; Zan, M. Driving Forces of the Changes in Vegetation Phenology in the Qinghai-Tibet Plateau. Remote Sens. 2021, 13, 4952.

Wang, X.; Xiao, J.; Li, X.; Cheng, G.; Ma, M.; Zhu, G.; Arain, M.A.; Black, T.A.; Jassal, R. S. No trends in spring and autumn phenology during the global warming hiatus. Nat Commun. 2019, 10, 1-10.

Yu, H.; Luedeling, E.; Xu, J. Winter and spring warming result in delayed spring phenology on the Tibetan Plateau. P Natl A Sci. 2010, 107, 22151-22156.